# High resolution microscopy to evaluate the efficiency of surface sterilization of Zea Mays seeds

Yalda Davoudpour*, Matthias Schmidt, Federica Calabrese, Hans Hermann Richnow, Niculina Musat

Department of Isotope Biogeochemistry, Helmholtz-Centre for Environmental Research (UFZ), Leipzig, Germany

* yalda.davoudpour@ufz.de

**Data Availability Statement:** All relevant data are within the manuscript.

**Funding:** Hans Hermann Richnow (HHR) received the fund from Deutsche Forschungsgemeinschaft (DFG) (Project No. RI 903/7-1) (https://www.dfg.

## Abstract

Surface sterilization of seeds is a key step in providing microorganisms-free seeds for numerous applications like understanding the role of seed-borne microorganisms in plant development, studying microbial cells-plant interactions by inoculating model microorganisms in a simplified system or selective cultivation of seed endobionts. However applying efficient treatment for surface sterilization of seeds without affecting the plant growth is not an easy task. In this study we aimed to provide an efficient surface sterilization treatment for maize seeds using i) hydrogen peroxide (HP), ii) sodium hypochlorite (SH) and iii) ethanol-sodium hypochlorite (EtOH-SH) under stirring (st) and vacuum-stirring (va-st) conditions. We used fluorescence microscopy and ultra-high resolution Helium Ion Microscopy (HIM) as powerful imaging approaches in combination with macroscopic techniques to visualize, quantify and evaluate the efficiency of seed sterilization, quality of root germination, seedlings and root hair development as well as the presence or absence of microorganisms on the root surface. Our results showed a strong reduction in microbial cell numbers of 4 orders of magnitude after the EtOH-SH treatments. Moreover, seeds exposed to EtOH-SH treatments displayed the lowest percentage of microbial growth (50%) and the highest percentage of germinated seeds (100%) compared to other sterilization treatments. HIM imaging proved the absence of microbial cells on the roots grown from seeds exposed to EtOH-SH treatments. Moreover, root hair development seemed not to be affected by any of the sterilization treatments. Our findings demonstrated that EtOH-SH treatments are significantly reducing the abundance of microbial cells from the surface of maize seeds and can be used with high confidence in future studies.

## Introduction

Seeds are the starting point of a growing plant and therefore seed-borne microorganisms are primary inoculum source of the plant associated microbial community [1]. However, the surface of seeds is normally colonized by microorganisms passed on from their mother plant (early colonizers) and via seed contact with fruit microorganisms (late colonizers) [1]. Another source of microorganisms colonizing seeds is soil in which the seeds are growing [2, 3].

de/). This project was carried out in the framework of the priority program 2089 "Rhizosphere spatiotemporal organization—a key to rhizosphere functions" funded by DFG. This work was supported by ProVIS Centre for Chemical Microscopy (established with funds provided by Europäischer Fonds für regionale Entwicklung (EFRE) und dem Freistaat Sachsen Program) at Helmholtz Centre for Environmental Research – UFZ for all authors.

**Competing interests:** Authors confirm that there is no competing of interests related to this publication.

To study the identity of microorganisms i.e. seed-borne vs non-seed borne, surface sterilization is commonly employed [4]. Removal of microorganisms from the surface of seeds is also important for the investigation of plant-microbe interactions in axenic and monoxenic model systems [5]. For selective cultivation of microbial endobionts of seeds an efficient surface sterilization might be also important. The surface sterilization is not an easy task because only microorganisms on the surface should be removed but plant cells must not be damaged by the procedure [6, 7]. In addition, the selection of sterilization agent and parameters are critical to obtain a proper plant growth, high germination rate with minimum microbial contamination and negative effect [8]. Various factors may affect the selection of sterilization procedure such as the type and the origin of the plant, and the level of microbial contamination, hence, it is not that one sterilization method fit all plants [9].

The widely applied chemicals for surface sterilization of seeds of various plants are hydrogen peroxide ($H_2O_2$), sodium hypochlorite (NaOCl), ethanol and mercuric chloride ($HgCl_2$). In the case of *Zea Mays* (maize) seeds, sterilization by NaOCl in concentration between 0.024% and 20% [10–15] up to 20 minutes have been reported. Surface sterilization of maize subjecting the seeds to 0.1% $H_2O_2$ [16] and 10% $H_2O_2$ [17] up to 30 minutes have also been proposed. An alternative treatment, with only 2 min exposure time of maize to 1% $HgCl_2$ solution has been reported [18]. After all these treatments the maize seeds germinated within 10 days. In addition, combinations of ethanol (mostly 70%) with NaOCl [19, 20] and $H_2O_2$ [17] with up to 5 min exposure have been applied. A previous study showed that increasing the sterilization time of maize using NaOCl from 0.5 to 5 hours under shaking, decrease not only the number of infected seeds but also germination rate [21]. The application of 1 min suction (vacuum) and 2.5 hours sterilization with shaking resulted in lower numbers of infected seeds [21]. It was reported that shaking during the procedure will increase mixing of sterilization solution with maize grains and suction will help to remove air bubbles from seeds and improve wettability of the seed surface [21].

Understanding the positive and negative effects of various sterilization agents on the removal of surface associated microorganisms, germination and plant development will help to select the best suited sterilization procedure for a specific species. The strong oxidizing characteristic of hypochlorites such as NaOCl is very effective in killing bacteria and reducing bacterial populations [22]. The high reactivity of hypochlorites with amines, amino acids, amides and nucleic acids lead to formation of $NH_4Cl$, aldehyde and $CO_2$ [23]. They are the typical products of the reaction between NaOCl and amino acids [23]. Diluted hypochlorite solutions forms hypochlorous acid (HOCl) which reacts particularly with organic amines and $NH_4$ [5]. It forms extremely toxic chloramines (a powerful oxidizing and very diffusible species), which can enter through cell membranes to react with components inside the cell for example DNA [5]. Ethanol is a strong sterilization agent which is very phytotoxic [22]. It is typically applied for only a few minutes or even seconds, generally at 70% concentration and prior to other sterilization agents (usually NaOCl) for increasing effectiveness [22, 24]. There are also some reports using Tween 20 as a surfactant in the sterilization solutions (usually together with NaOCl) to improve wettability of the seeds in order to increase efficiency of sterilization process [9, 24]. Since $HgCl_2$ is a highly toxic chemical and difficult to dispose, it has not been utilized broadly for maize surface sterilization [25]. $H_2O_2$ is a reactive component with a dual role of signaling (beneficial) and damaging (deleterious) for physiological properties and development of plants [26]. At low concentration $H_2O_2$ possesses fungicidal and germicidal activities without affecting seed growth and germination [27]. Enzymes available in many plant cells such as catalases and peroxidases decompose $H_2O_2$ into oxygen and water and protect cells from damaging effects of peroxides [27]. It was reported that after sterilization using 10% $H_2O_2$ no damage on maize plants was found [17]. However, a reducing of germination of

*Rhododendron wardii* seeds due to tissue damage by increasing the sterilization time from 10 to 20 min and $H_2O_2$ concentration from 20 to 30% was reported [27]. Hence, the damaging effect of $H_2O_2$ on plant tissues and growth after seed sterilization can be different from species to species and should be empirically determined [28].

For analyzing the efficiency of seed sterilization, the blotter technique (filter paper) and potato dextrose agar (PDA, agar plate) test have been widely used to detect seed-borne fungi and bacteria [6, 29–31]. Agar plate is used as growth media for seed associated microorganisms and hence for the identification and discarding of seedlings showing microbial growth [6, 29]. For analyzing the root development of maize following sterilization, previous studies mainly utilized filter paper [10, 12, 15, 20] and only few applications of PDA have been reported [14, 17, 21].

Here, we used maize as model species to study the sterilization efficiency of different surface sterilization agents namely $H_2O_2$, NaOCl and ethanol-NaOCl using stirring and vacuum-stirring for the removal of the seeds' surface associated microorganisms such as bacteria and fungi and to investigate the effect of these agents on germination and root growth. We evaluated the efficiency of sterilization processes quantitatively and qualitatively by a multi-scale approach combining macroscopic observations with microscopic approaches such as fluorescence microscopy and ultra-high resolution Helium Ion Microscopy (HIM). Our findings showed that ethanol-NaOCl based treatment is the best suited treatment for the surface sterilization of maize seeds.

## Materials and methods

### Maize seeds

Wild type (WT) maize seeds were provided by the Institute of Crop Science and Resource Conservation, University of Bonn. Stock solutions of 12% NaOCl, 30% $H_2O_2$, Tween 20 (Carl Roth GmbH, Germany) and 100% ethanol were applied to prepare sterilization agents. The whole procedure was conducted using sterilized containers and solutions under laminar flow hood. The experimental design is illustrated in Fig 1.

### Surface sterilization procedures

A total number of 72 seeds were surface sterilized using three treatments either by mild stirring (st) or using mild vacuum (800 mbars absolute pressure)-mild stirring (va-st). In the first treatment, surface sterilization of seeds was performed using 10% $H_2O_2$ for 10min. Then the solution was decanted and seeds were rinsed in water for 2min. The procedure was repeated 4 more times (HP treatment). In the second sterilization treatment, a similar procedure was conducted using 70 mL of 10% NaOCl solution which was mixed with 17 μl of Tween 20 (SH treatment). In the third sterilization treatment, seeds were pretreated first with 70% ethanol for 3min. The ethanol was decanted, and seeds were sterilized two times by a mixture of 10% NaOCl and 17 μl of Tween 20, each for 15min. Finally the seeds were washed five times using water, each for 2 min (EtOH-SH treatment). From now on we will apply these abbreviations (mentioned in Table 1) all over the manuscript for samples identification. Each experiment was conducted twice, independently and each treatment was applied on 2 individual seeds.

### Preparing sterilized seeds for germination and sterilization efficiency tests

In order to analyze the efficiency of sterilization, seeds were divided into three groups. 12 seeds were placed into petri dishes with 5ml of potato dextrose broth (PDB) [32] medium to evaluate the turbidity and microbial cell abundance, 12 seeds were placed in petri dishes with a

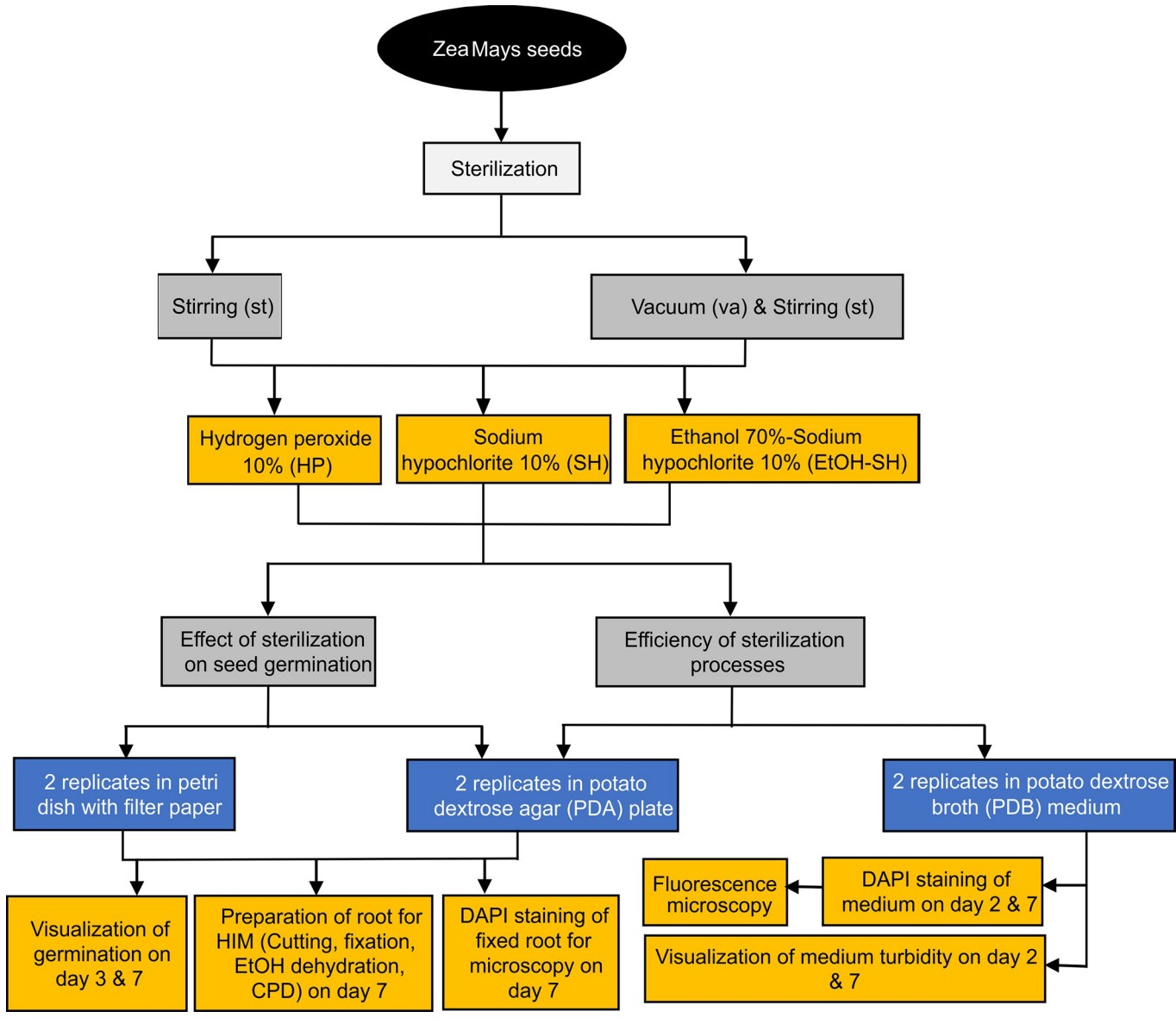

**Fig 1.** Experimental design and the applied methods.

**Table 1. Sterilization treatments and the corresponding abbreviations.**

| Sterilization treatment | Abbreviation |
|---|---|
| $H_2O_2$ with vacuum and stirring | HP-va-st |
| $H_2O_2$ with stirring | HP-st |
| NaOCl with vacuum and stirring | SH-va-st |
| NaOCl with stirring | SH-st |
| Ethanol-NaOCl with vacuum and stirring | EtOH-SH-va-st |
| Ethanol-NaOCl with stirring | EtOH-SH-st |
| Untreated seeds | Ref |
| Control medium | Con. Med |

moistened filter paper to study the germination of seeds after each sterilization procedure and 12 seeds were placed in PDA [32] plates to observe the microbial growth on seeds after each sterilization treatments. The petri dishes with PDB were incubated in an incubator in darkness at 28±2°C for up to 7 days. As control, the sterile PDB medium (Con. Med) was selected for turbidity visualization and comparison with the PDB medium of surface sterilized seeds. The petri dishes for germination test and PDA plates were rapped with sterile aluminum foil and placed in darkness at 26°C for up to 7 days. Similar procedures were performed for the preparation of untreated reference sample (Ref, seeds without sterilization).

## Evaluating number of germinated seeds vs. number of seeds with microbial growth

The evaluation of germination after 3 and 7 days was performed by capturing image from the seeds and their roots in petri dishes. Similar approach was conducted for the comparison of seeds microbial growth in PDA plates. Additionally, we calculated the effect of various sterilizations on the number of seeds showing microbial growth and those that germinated after 7 days using the Eqs 1 and 2 [33].

$$Seeds\ with\ microbial\ growth\ (\%) = \left(\frac{Number\ of\ seeds\ with\ microbial\ growth}{Total\ number\ of\ seeds}\right) \times 100 \quad (1)$$

$$Germinated\ seeds\ (\%) = \left(\frac{Number\ of\ grown\ seeds}{Total\ number\ of\ seeds}\right) \times 100 \quad (2)$$

## Analysis of PDB medium

To check the PDB medium where the sterilized seeds were deployed and to verify the sterilization efficiency, we applied two procedures. First, we compared the turbidity of liquid PDB medium macroscopically using a photo camera. Second, we sampled 700 µl of non-turbid PDB medium and 70 µl of turbid PDB medium after 2 and 7 days of incubation and filtered these volumes on Au/Pd (80/20) coated filters (GTTP type, 0.2 µm pore size PC membrane, 25 mm diameter, Merck Millipore, Germany) using a sterile multichannel filtering device (Millipore, Germany). Following filtration, filters were dehydrated by 30, 50, 70 and 80% ethanol and air dried. Filter pieces were stained with 50 µl DAPI solution (1 µg/mL) in dark for 10 min, washed with ultrapure Milli-Q (MQ) water, dipped in 80% ethanol and air dried. For microscopy, the filters were embedded in 20 µl of Citiflour (Science Services GmbH, Germany): Vectashield (L I N A R IS Biologische Produkte GmbH, Germany) (CV) mixture (1:4 vol/vol) on a glass slide covered by a glass cover slip and stored at –20°C prior to visualization. The filters were imaged using a fluorescent microscope (DAPI filter, 100X, numerical aperture N:A 1.4 oil objective, Imager. Z2, Zeiss, Germany) to count the microorganisms growing in the PDB medium after various sterilization treatments. Two independent filter pieces from two independent petri dishes of the same treatment were analyzed (imaged and counted). In order to determine the total DAPI stained cells per mL of PDB medium we applied the following steps: i) counting the DAPI stained cells per area of the field of view (image), average the counts obtained for all fields of view of the same filter piece and average counts from duplicate filter pieces belonging to the same sample to obtain a single value. The area of each picture was constant as we used always for imaging 100X objective and was equal to 6005.7 $\mu m^2$; ii) calculating the counts for the total filter area (DAPI cells per picture * filter area)/picture area). Area of the filter was constant as we used 25 mm diameter filters where the r of the filtration zone was always 9.5 mm; iii) knowing the volume filtered on each filter (always constant for turbid

samples 0.07 mL and for non-turbid samples 0.7 mL) we calculated DAPI stained cells per mL of PDB medium. We counted two replicate filters for each treatment. Cell counting was performed automatically using ImageJ-win64 software and manually for high confidence. In the case of samples with high number of microbial cells, 3–5 fields of view (each of 89.53 μm×67.08 μm) summing up to 1000 cells were randomly imaged and counted. For samples having lower cell numbers, 150 fields of view (each of 89.53×67.08 $\mu m^2$) per filter (randomly selected) were imaged and counted. DAPI cell counts from different fields of view of the same sample were averaged and the final value was used to calculate total DAPI cells $mL^{-1}$ for each treatments. For good counting statistics, DAPI counting was done on two duplicate filters from each replicate and each individual treatment.

## Analysis of roots using fluorescence microscope

To analyze if after various sterilization procedures microorganisms are still present on the surface of the newly developed roots, we used DAPI staining and fluorescence microscopy. After 7 days of seeds incubation, roots were cut to the length of approximately 1 cm, fixed using 2% paraformaldehyde (PFA) in 1X phosphate-buffered saline (PBS) overnight at 4˚C. After fixation, the roots were rinsed with sterile MQ water to remove fixative and air dried on a filter paper. 80 μl of DAPI (1 $\mu gmL^{-1}$) was added on each root fragment to completely cover it and incubated for 30min in dark followed by gentle washing using MQ water, air drying, embedding in CV and imaging under fluorescence microscope.

## Analysis of roots by Helium Ion Microscope (HIM)

High resolution imaging of microorganisms on the surface of roots was done using a Zeiss Orion NanoFab Helium Ion Microscope (HIM) (Carl Zeiss Microscopy, Peabody, MA). After 7 days of root growth, for all treatments, roots of 1 cm length developed on both PDA and filter paper, were cut, fixed by 2% PFA in 1X PBS overnight at 4˚C and dehydrated using an ethanol series (from 30% to 100%, in 10% steps). Then the roots were dried using critical point drying (CPD) machine following the manufacturer recommendations (EM CPD 300, Leica, Austria). CPD was used to preserve the structure of root during drying. Among various drying methods, the application of CPD for the preparation of undamaged roots for scanning electron microscopy (SEM) analysis has been reported [34, 35]. For imaging, the roots were placed onto stubs as used in SEM, fixed using an epoxy glue and analyzed by HIM. 2 root fragments corresponding to PDA and filter paper, per treatment were randomly selected for HIM analysis. Initially, fields of view of 1100X1100 μm were scanned followed by imaging of randomly picked smaller fields of view of 20X20 μm within the selected large fields.

# Results and discussion

## Macroscopic investigation of surface sterilization treatments

The influence of various surface sterilization treatments on seed germination was investigated by growing the seeds, after sterilization, on watered filter paper (blotter method) [31]. After 3 days, we observed germination of all seeds from all different sterilization treatments (Fig 2A–2F) and the Ref sample (Fig 2G), however, not all replicates germinated properly (Fig 2A–2G). After 7 days, seedling development from all replicates and treatments was clearly observed (Fig 2H–2N). Over all, there was not a significant difference in seedling development between the untreated reference seeds and the treated ones after 7 days (Fig 2H–2N), suggesting that the applied sterilization treatments do not influence the germination and seedling development.

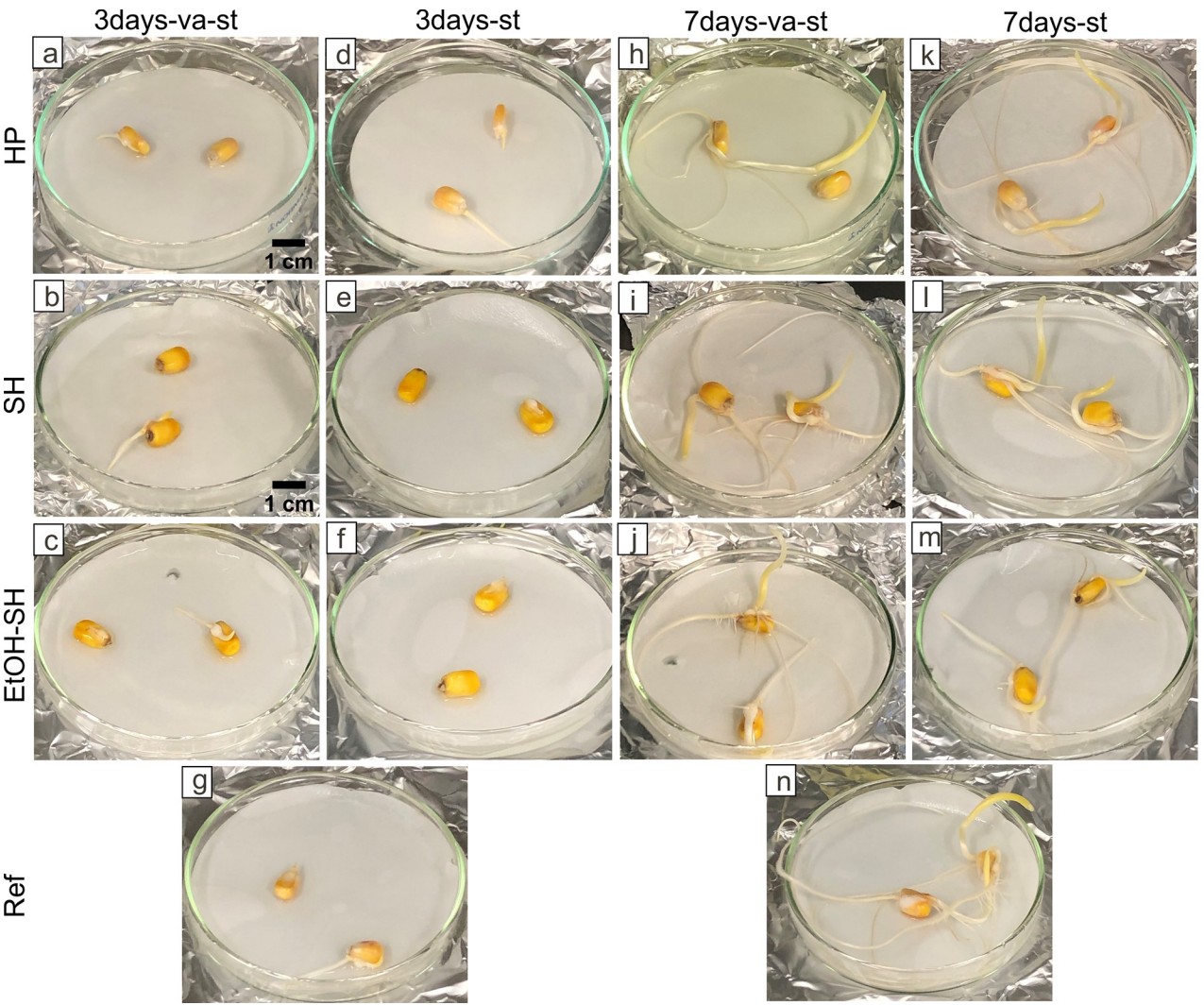

**Fig 2. Seeds germination and seedling development after exposure to different sterilization treatments.** Images depicting seeds exposed to different sterilization treatments and grown on filter paper for 3 days (a-f) and 7 days (h-m) as well as untreated seeds (g, n). Scale bar in (a) is applicable for (h, g, n). Scale bar in (b) is applicable for (c, d, e, f, i, j, k, l, m).

No significant difference was observed between treatments using vacuum-stirring and only stirring on seed germination (3 days) and seedling development (7 days) (Fig 2).

In order to check the removal of seed associated microorganisms after sterilization treatments we applied the PDA plate method [30]. After 3 days, first sign of microbial growth was clearly observed in the Ref sample (Fig 3G) and slightly in HP-va-st, HP-st and SH-st treated samples (Fig 3A, 3D and 3E, black arrows) while the others remained sterile (Fig 3B, 3C and 3F). After 7 days, the highest microbial growth was observed, as expected, in the Ref sample (Fig 3N). In the sterilization treatments the following trend was observed: HP-va-st and HP-st having the highest microbial growth followed by SH-va-st and SH-st with moderate growths and EtOH-SH-va-st and EtOH-SH-st treatments with very little microbial growth (Fig 3H–3M). An exception was observed by the EtOH-SH-st sample where only one germinated seed showed microbial growth while the other one remained sterile (Fig 3M). This result suggests that the EtOH-SH treatment is efficient in removing seed associated microorganisms.

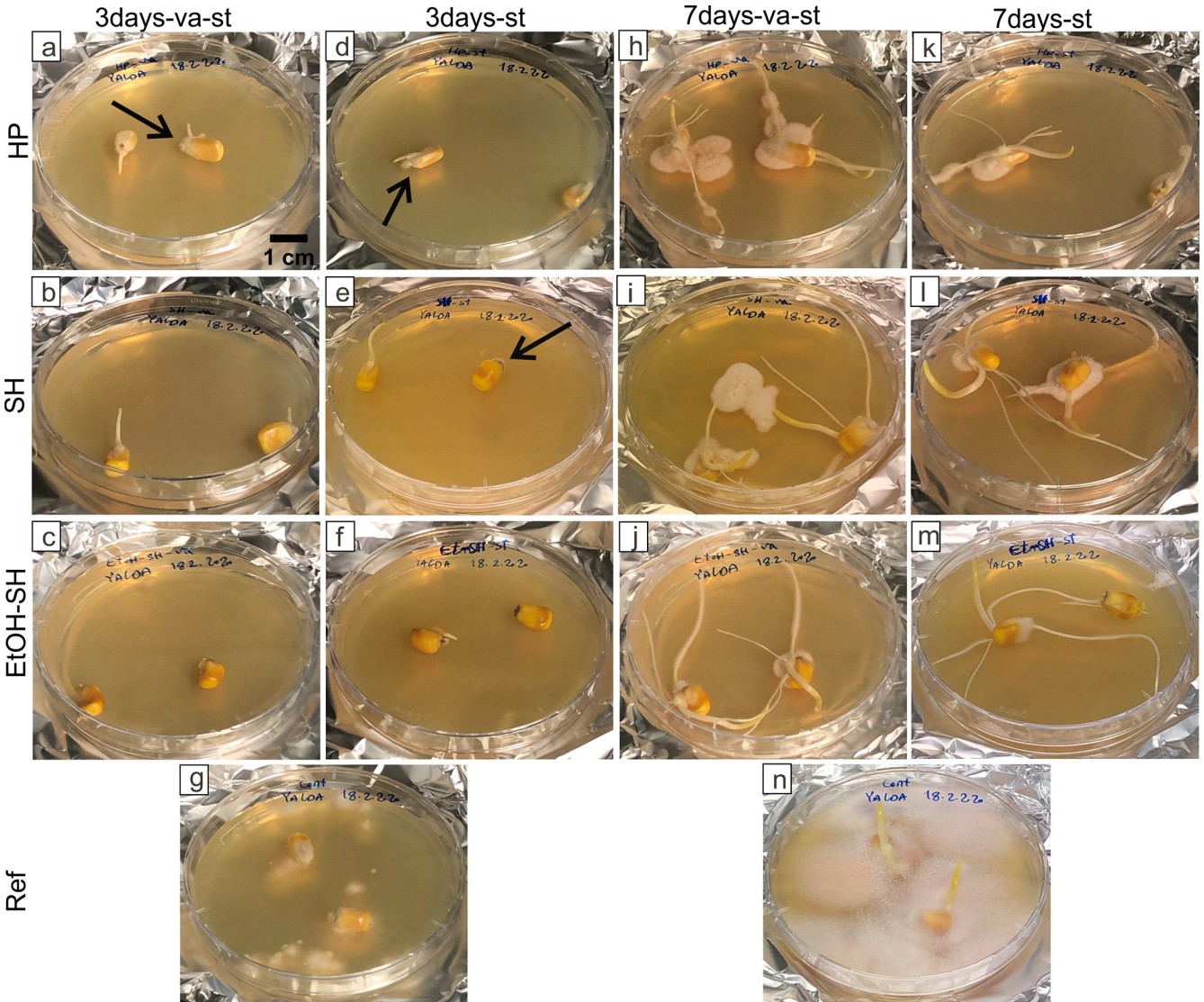

**Fig 3. Microbial growth on seeds exposed to different sterilization treatments.** Macroscopic images showing microbial growth on seeds exposed to sterilization treatments and grown on PDA for 3 days (a-f) and 7 days (h-m) as well as untreated seeds (g, n). Black arrows show first sign of microbial growth after 3 days. Scale bar in (a) is applicable for all panels.

Microbial growth visually observed on PDA plates after SH and EtOH-SH surface sterilizations can be mainly attributed to seed-borne microorganisms because we did not observe the microbial growth after 3 days in these samples. Similar results have been reported about seed-borne fungi in maize after sterilization using SH [36] or EtOH-SH [6]. However, in the case of HP sample microbial growth was visibly faster, only after 3 days of incubation and it can be due to both inefficient sterilization and seed-borne microorganisms.

The effect of sterilization on germination and the efficiency of seeds sterilization in removing of surface-associated microorganisms were calculated for all 72 seeds (Fig 4). These results point out that the EtOH-SH and SH are more efficient treatments for seed sterilization and the EtOH-SH treatment led to the highest percentage of seed germination (100%). However none of the sterilization treatments could completely remove all seed associated microorganisms which is consistent with previously reported studies on maize seeds [36]. SH and EtOH-SH

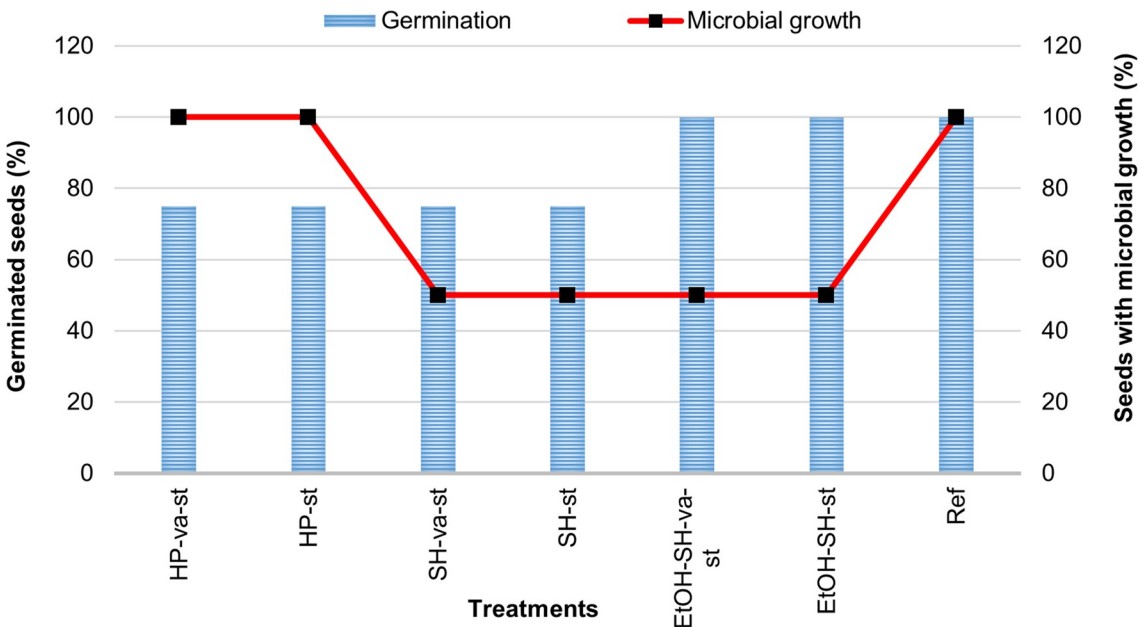

**Fig 4. Seeds germination vs microbial growth after sterilization treatments.** Germination (blue bars) and microbial growth (red line-dot) depicted as % and calculated after 7 days growth on filter paper (for germination) and on PDA (for microbial growth) of seeds sterilized by different treatments.

treatments could reduce the number of seeds showing microbial growth (to 50%) in comparison with other treatments (100%). A possible explanation, could be the formation of oxidizing chloramines from SH that can react with cell components via entering cell membrane [5] and prevent microbial growth. Previous studies have also reported a reduction of seeds' with microbial growth when EtOH rinsing was used before SH treatment [6] or simply just by applying SH treatment [33] while HP treatment was ineffective for seeds sterilization [5]. The observed inefficient HP sterilization results is ascribed either to the elimination of HP by washing or readily diffusion of HP to cells and decomposition by catalase which in both scenarios it is not powerful for the sterilization [5].

## Qualitative and quantitative evaluation of sterilization efficiency

In order to investigate the sterilization efficiency qualitatively and quantitatively, treated seeds were incubated for 2 as well as 7 days into PDB liquid medium. Microbial growth and cell abundance were investigated macroscopically following turbidity development (Fig 5) and microscopically by DAPI staining and fluorescence microscopy (Figs 6–9). The turbidity of PDB liquid medium can be utilized as a quick and qualitative indicator for the growth of microorganisms [37, 38]. Visual comparison was done against PDB medium named control medium (Con. Med), and the PDB containing untreated seeds (fully turbid), which were incubated in the same conditions as treated samples (Fig 5).

After 2 days of seeds incubation, we observed turbid PDB medium in both replicates of HP-st (Fig 5B), one replicate of HP-va-st (Fig 5A) and the Ref sample (Fig 5E), while none of the replicates of SH and EtOH-SH treatments showed any turbidity (Fig 5A and 5B). However, turbidity was observed in both replicates of HP-va-st, SH-va-st and HP-st as well as one replicate of EtOH-SH-st sample after 7 days of seeds incubation suggesting microbial growth (Fig 5C and 5D). The rest of the treated samples (EtOH-SH-va-st and SH-st) (Fig 5C and 5D) and Con. Med (Fig 5H) showed no turbidity after 7 days. These findings are consistent with the

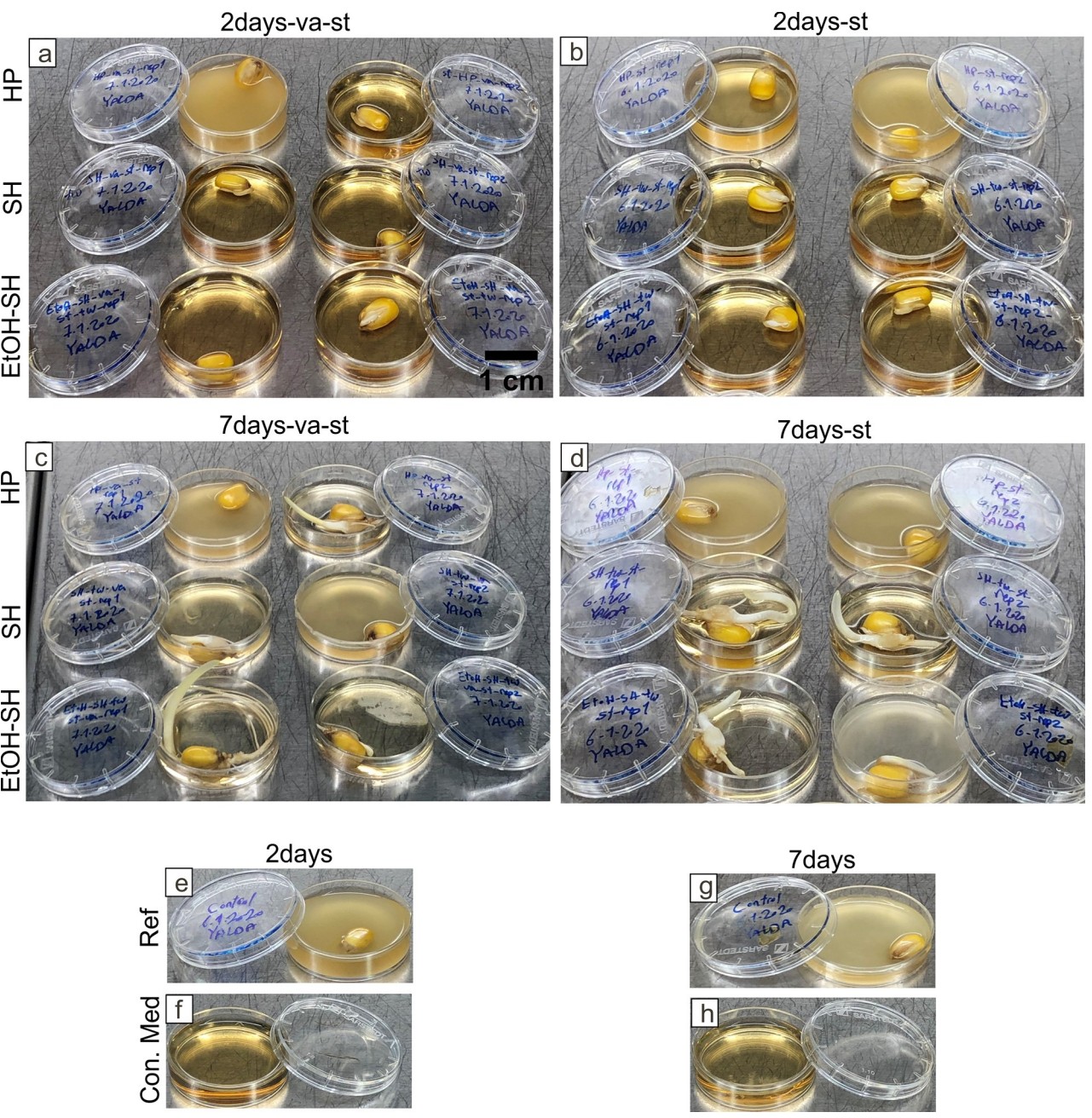

**Fig 5. Turbidity of the PDB medium as indicator for microbial growth over time.** Images show sterilized seeds by various treatments incubated in PDB medium for 2 days (a, b) and 7 days (c, d) as well as untreated seeds (e, g) and control medium (f, h). The corresponding turbidity development suggesting microbial growth. Scale bar in (a) is applicable for all panels.

results obtained from the comparison of sterilization efficiency test performed using PDA plate method (Fig 3). Various behavior of seeds from same batch to the same sterilization treatment has been previously reported [39]. A possible reason why only one seed replicate is efficiently sterilized while the other replicate is not, may be related to the different initial load of associated microorganisms prior to sterilization treatments [40]. These results also confirm the more efficient sterilization by SH and EtOH-SH treatments for removing of the surface associated microorganisms from maize seeds compared to HP treatment.

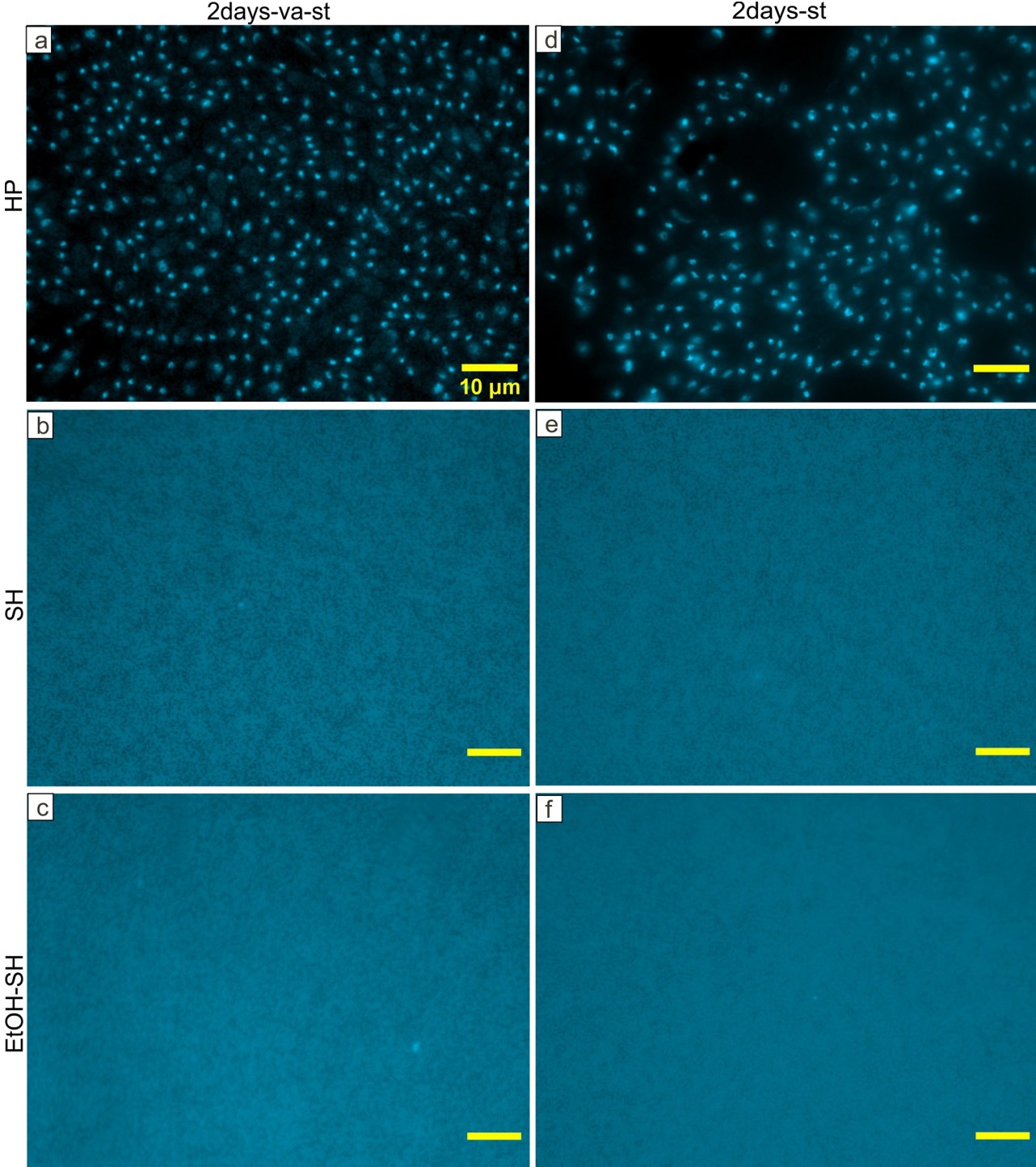

**Fig 6. Representative fluorescence microscopy images of DAPI stained PDB medium filtrate after 2 days of incubation.** Fluorescence microscopy micrographs showing high abundance of microbial cells after HP treatments (a, d) and very low number of microbial cells after SH (b, e) and EtOH-SH treatments (c, f). Scale bar represents 10μm for all images.

Microscopic investigation of filters containing PDB medium and seeds after 2 days of incubation showed the lowest abundance of DAPI cells in the range of $10^3$ to $10^4$ cells mL$^{-1}$ for all

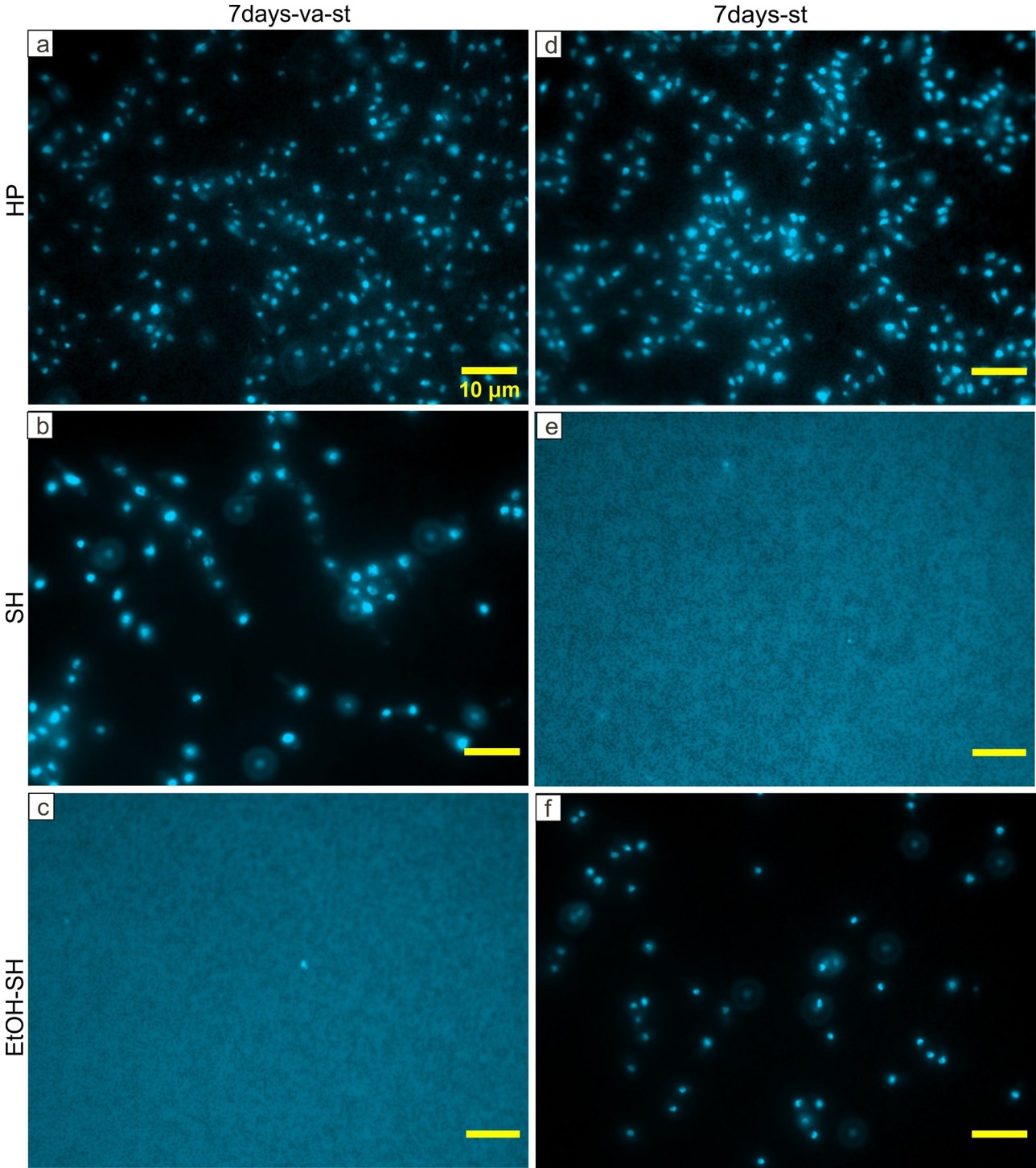

**Fig 7. Representative fluorescence microscopy images of DAPI stained PDB medium filtrate after 7 days of incubation.** Fluorescence microscopy micrographs showing high abundance of microbial cells after HP treatments (a, d), SH-va-st (b) and EtOH-SH-st (f) and very low abundance of microbial cells after SH-st (e) and EtOH-SH-va-st (c) treatments. Scale bar represents 10μm for all images.

SH and EtOH-SH treatments (Figs 6B, 6C, 6E, 6F and 9A). In contrast, the HP treatment and untreated seeds illustrated cell numbers in the range of $10^7$ to $10^8$ cells mL$^{-1}$ (Figs 6A, 6D, 8A

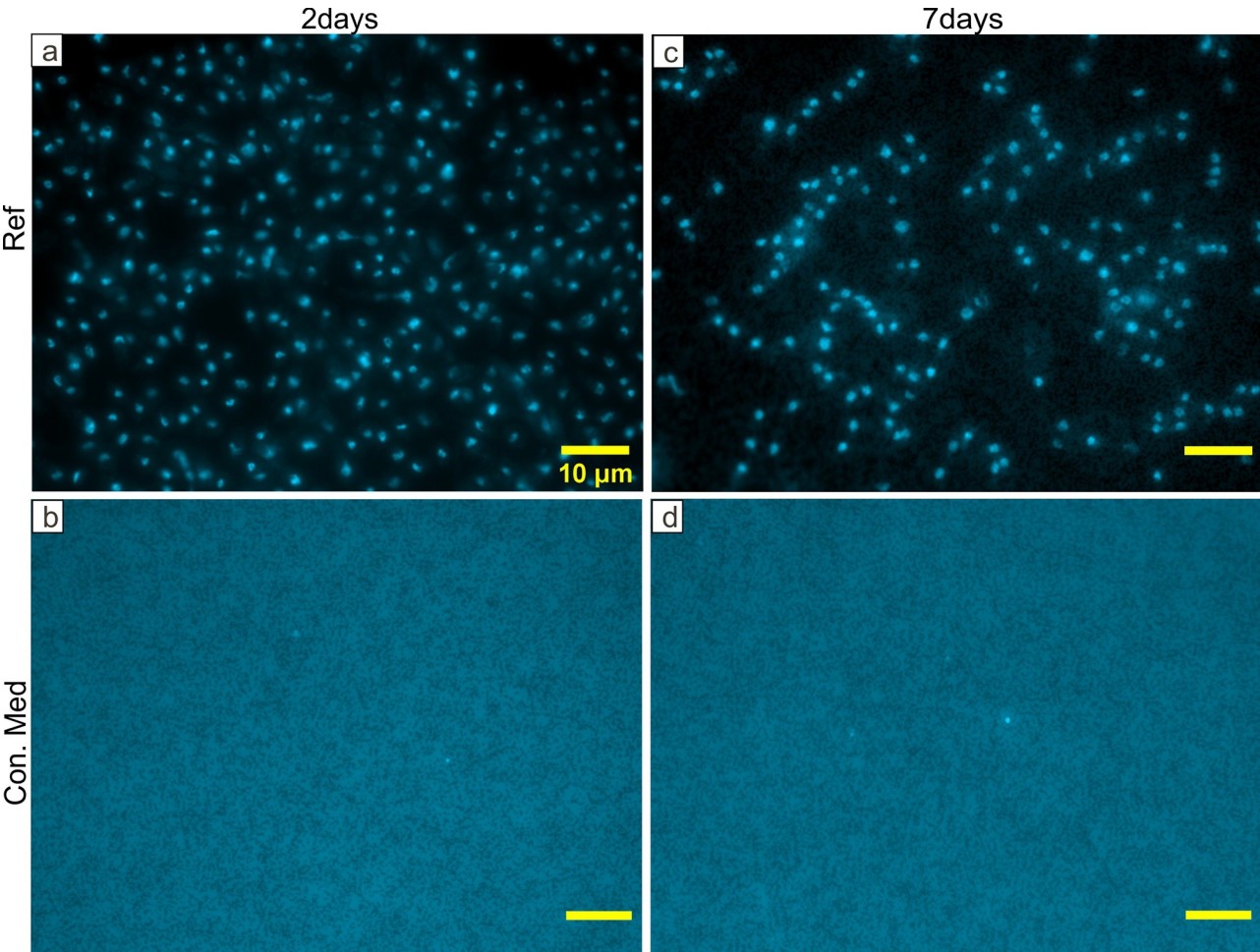

**Fig 8. Representative fluorescence microscopy images of DAPI stained PDB medium of untreated seeds and control medium.** Fluorescence microscopy micrographs showing high abundance of microbial cells after 2 days (a) and 7 days (c) incubations of the untreated seeds and control medium without seeds (b, d) incubated in the same conditions. Scale bar represents 10μm for all images.

and 9A). After 7 days of incubation, only two treatments of SH-st and EtOH-SH-va-st kept low cell numbers in the range of $10^4$ cells mL$^{-1}$, comparable with those of 2 days incubation suggesting that the cells have lost their viability after these sterilization treatments (Figs 7C, 7E and 9B). The SH-va-st and EtOH-SH-st treatments showed an increase in cell numbers from $10^3$ to $10^7$ cells mL$^{-1}$ either in one or in both replicates (Figs 7B, 7F and 9B). Regarding the HP treatments and the Ref sample the cell numbers stayed relatively similar with those recorded at 2 days incubation (Figs 7A, 7D, 8C and 9B). Based on DAPI cell counting alone, SH-st and EtOH-SH-va-st treatments seem to be the most efficient treatments for seeds surface sterilization. However, when combined with macroscopic analyses of microbial growth on PDA plates (Fig 3) and PDB turbidity evaluation (Fig 5), the most efficient sterilization treatment is EtOH-SH-va-st.

## Analysis of roots using fluorescence and Helium Ion Microscopy

Imaging of DAPI stained root fragments emerged from seeds exposed to different sterilization treatments and grown for 7 days showed the presence of microbial cells on all analyzed roots (Fig 10). We cannot ascertain if the presence of microorganisms on the root fragments

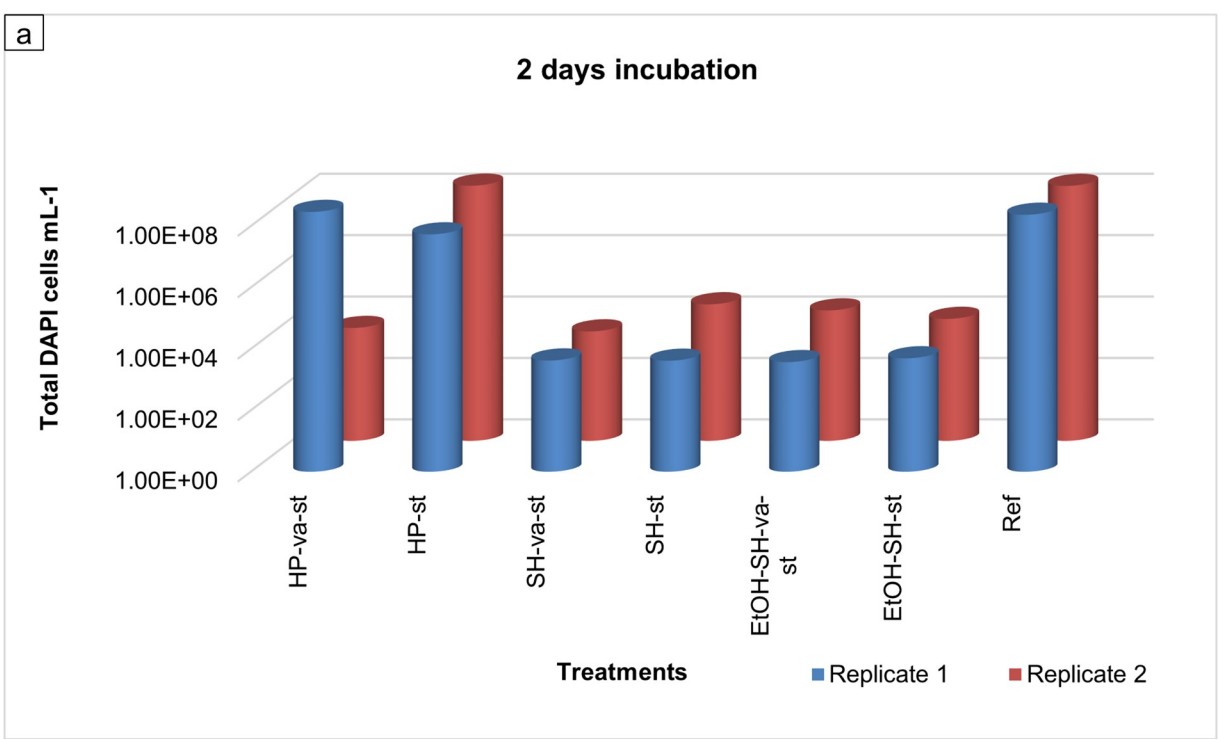

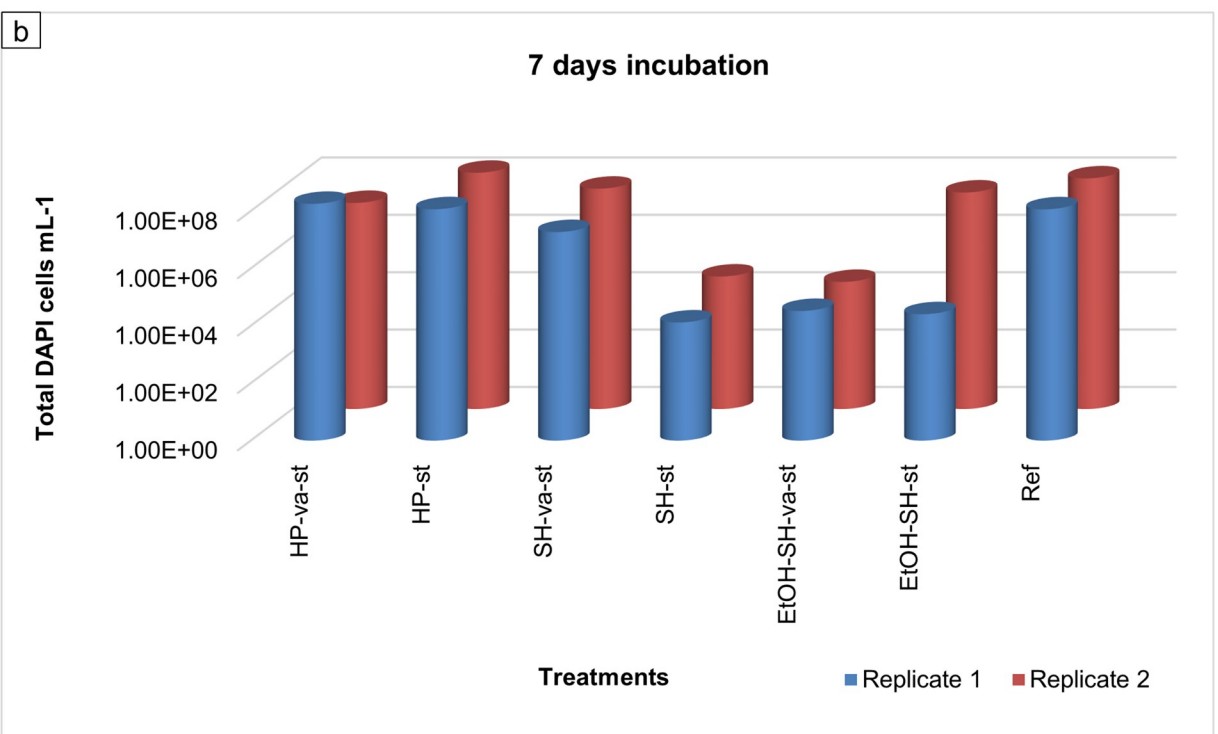

**Fig 9. Graphical representation of total DAPI counts for various sterilization treatments.**

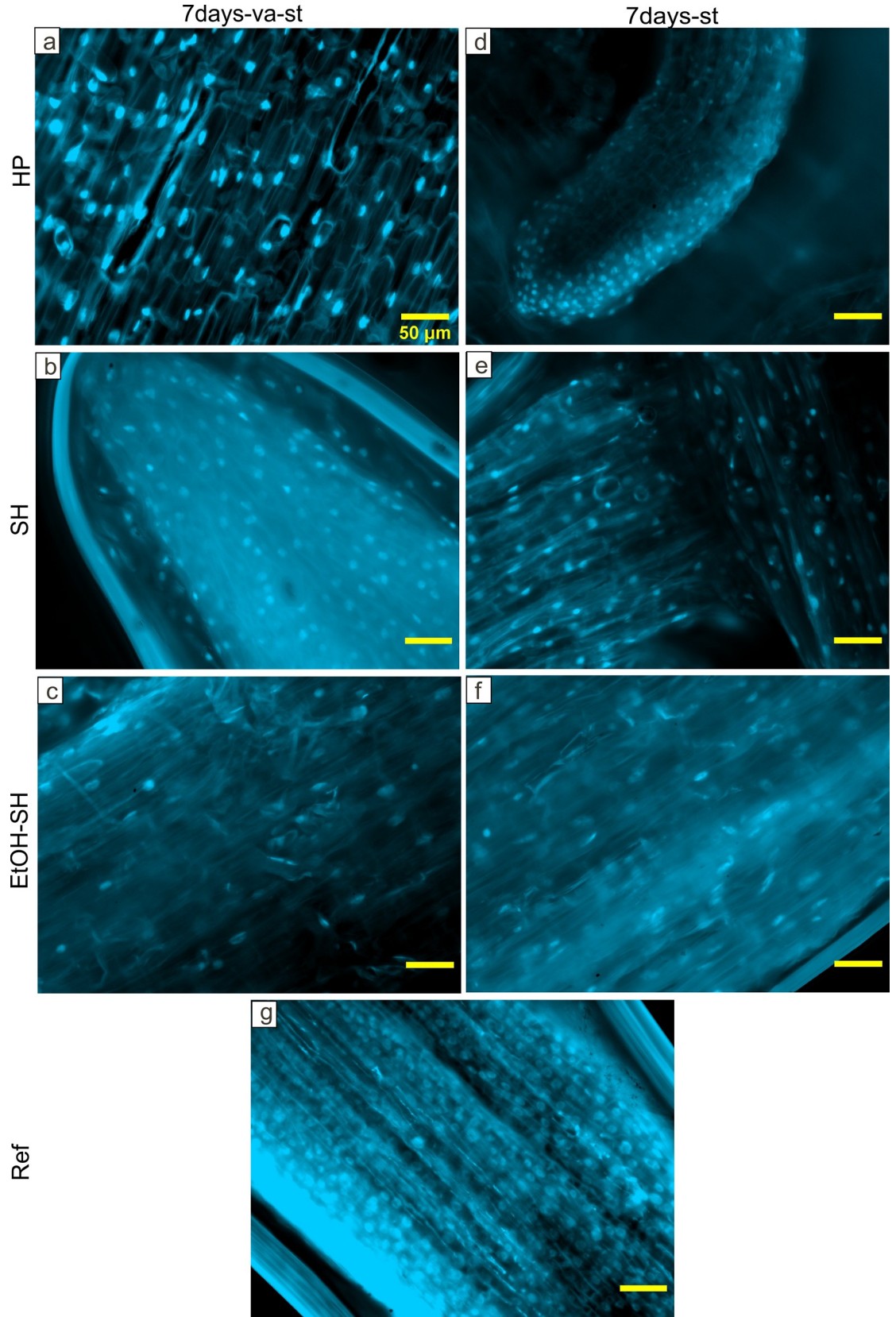

**Fig 10. Selected fluorescence microscopy images of DAPI stained roots after 7 days of root development.** Root fragments developed from seeds exposed to HP, SH and EtOH-SH sterilization treatments showing the presence of microbial cells. Scale bar 50μm applies to all images.

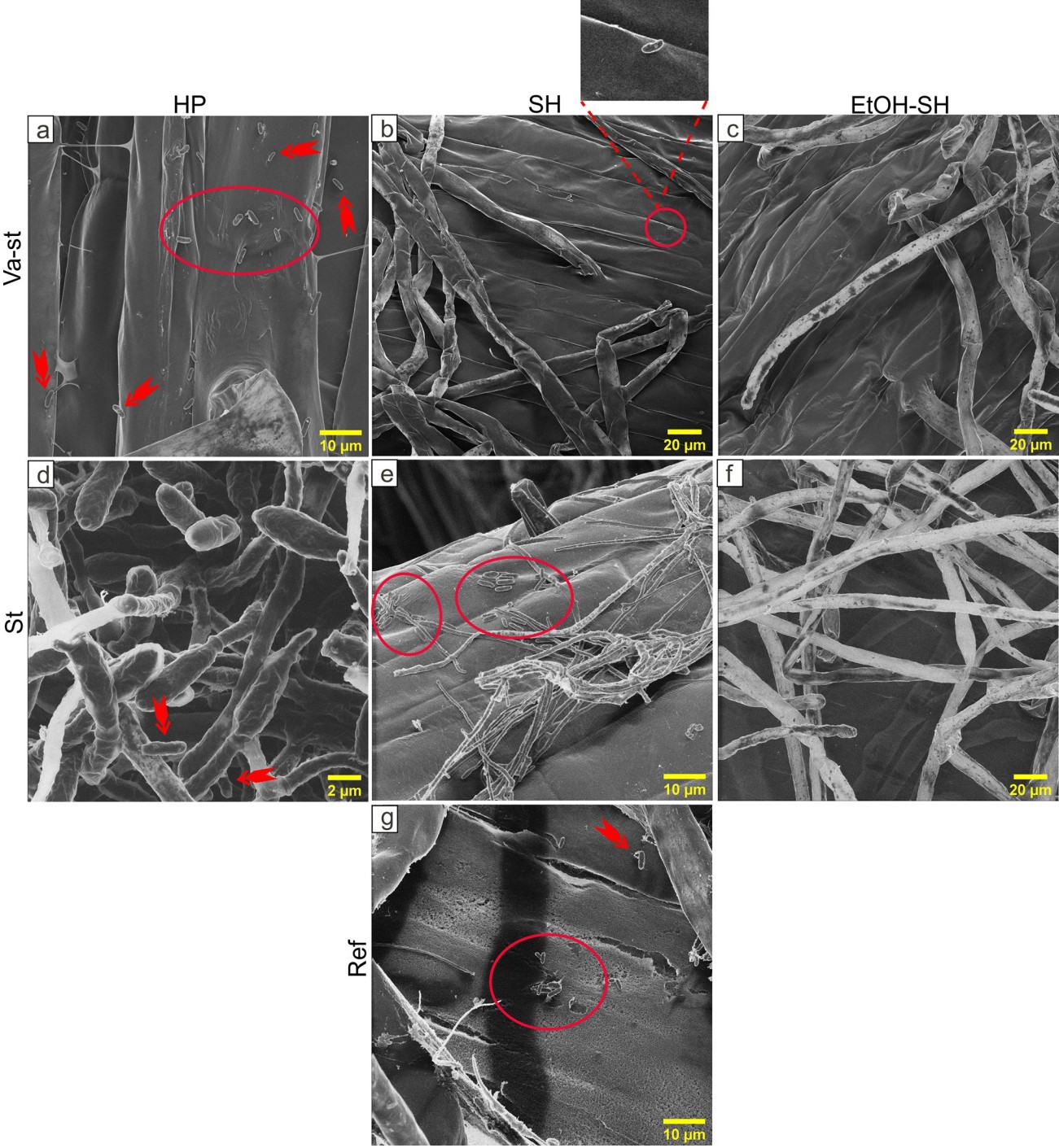

**Fig 11. Representative HIM images showing microbial cells on root surface after different sterilization treatments.** HIM micrographs show the presence of microorganisms (red arrows and circles) on roots grown from seeds exposed to HP (a, d) and SH (b, e) similar with the untreated seeds (g) while no microbial cells were observed after EtOH-SH treatments (c, f).

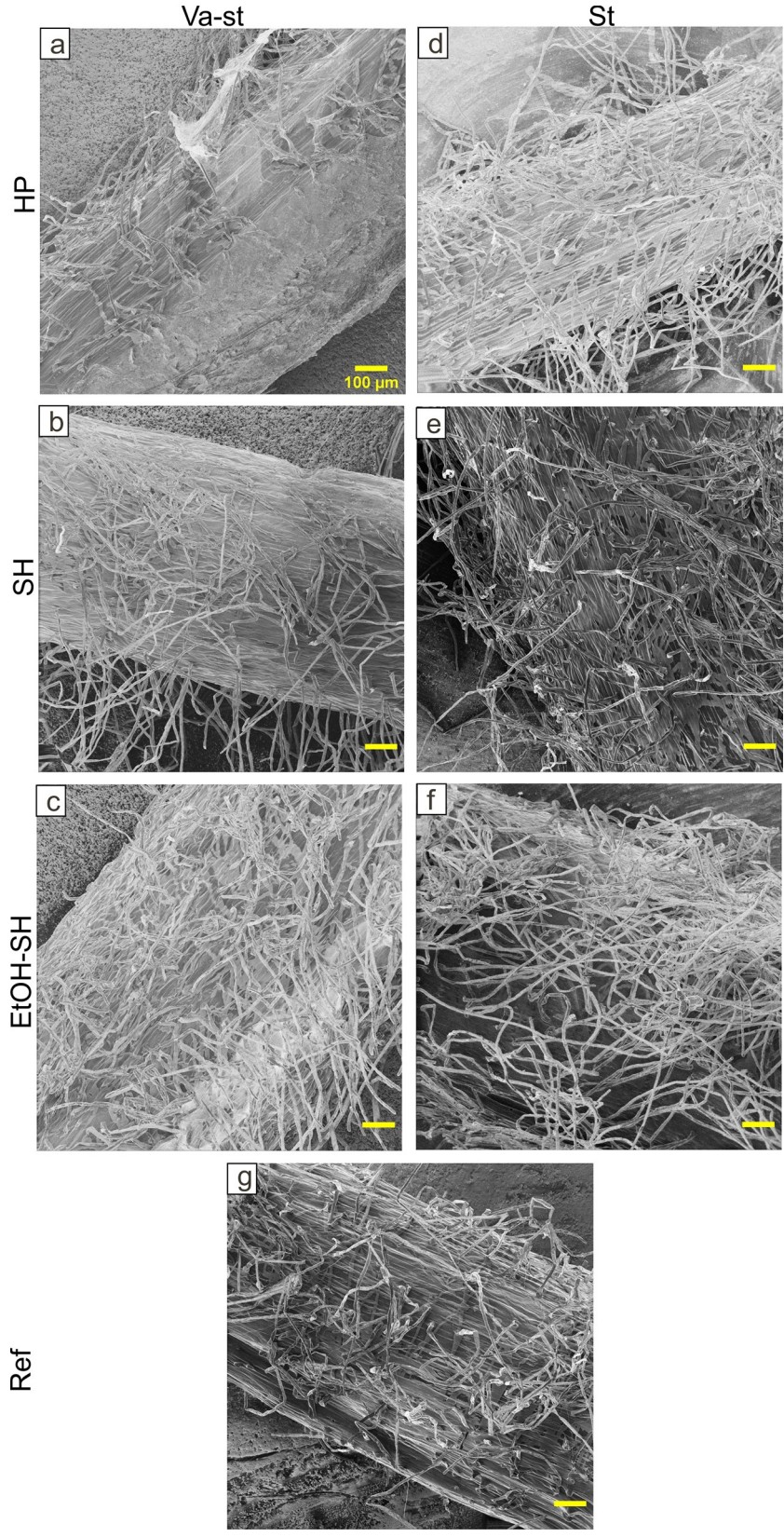

**Fig 12. Representative HIM images showing root hair development after different sterilization treatments.** HIM images showing no apparent negative effect on root hair development after sterilization treatments. Scale bar 100μm applies to all images.

occurred due to inefficient sterilization of seeds or to the seed-borne microorganisms which migrated to the root during the germination [29, 41]. We further investigated the surface of newly emerged roots by HIM.

Using HIM we were able to detect microorganisms on the root surface in all treatments except EtOH-SH-va-st and EtOH-SH-st. In these treatments, analyzed root surfaces seemed to be virtually free of microorganisms (Fig 11C and 11F). The HP-va-st, SH-va-st, HP-st and SH-st treated roots contained bacteria-like cells (mostly rod shaped) (Fig 11A, 11B, 11D and 11E). Bacteria-like cells were observed all over the surface of these samples and close to the junction of epidermal surface as single cells and in agglomerations. Furthermore, long filaments (up to 2 μm wide) similar to hyphae of fungi, were observed especially in HP-st and SH-st samples (Fig 11D and 11E). It has also been reported that generally diameter of root hairs (30μm) are larger than those of mycorrhizal fungal hyphae (10μm while 2μm for tip hyphae) [42]. The presence of fungi on the root surface has previously been observed by SEM imaging of Arabidopsis roots [43]. The diameter of the root hairs was measured in the HIM micrographs to about 8–18μm which is consistent with the reported diameter of root hair for maize [44, 45]. Moreover, according to HIM images of roots fragments it seems that the applied sterilization treatments had not a negative impact on the root hair development (Fig 12A–12G). Root hairs are formed from root epidermal cells and are very important for relatively immobile nutrients uptake by increasing the absorption surface area [46]. In maize any epidermal cell can create randomly a root hair [47]. A previously reported overview SEM image of maize root with root hairs developed from SH sterilized seeds [48] showed similar results to those obtained from SH treatment in this study.

## Conclusions

The quantitative and qualitative results of our study showed that EtOH-SH is the most efficient treatment for maize seed sterilization, drastically reducing the number of microbial cells on seeds surfaces, up to four orders of magnitude in comparison with HP and SH treatments. In addition, EtOH-SH seed sterilization treatment led to reduced microbial load or microorganisms-free surfaces of newly grown roots. Our data suggest that the EtOH-SH treatment, on one hand, lead to lysis of microbial cells hence strongly reducing their numbers and, on the other hand, the remaining cells seem not to be able to divide and grow anymore, even after 7 days of incubation time. None of the surface sterilization treatments seem to negatively influence the root hairs development when compared to the untreated seeds. Overall, our study provides an evidence for an efficient sterilization treatment of maize seeds which can be used with confidence by further studies.

## Acknowledgments

The authors acknowledge the support of Deutsche Forschungsgemeinschaft (DFG) for the Integration of Refugee Scientists and Academics for Dr. Yalda Davoudpour. Seeds of the maize wild type (WT) were provided by Caroline Marcon and Frank Hochholdinger (University of Bonn) and authors are thankful for it. The authors would also like to thank Katja Nerlich for technical support.

## Author Contributions

**Conceptualization:** Hans Hermann Richnow, Niculina Musat.

**Data curation:** Yalda Davoudpour, Matthias Schmidt.

**Formal analysis:** Yalda Davoudpour, Matthias Schmidt, Federica Calabrese.

**Investigation:** Yalda Davoudpour.

**Methodology:** Yalda Davoudpour, Niculina Musat.

**Supervision:** Hans Hermann Richnow, Niculina Musat.

**Visualization:** Yalda Davoudpour.

**Writing – original draft:** Yalda Davoudpour.

**Writing – review & editing:** Yalda Davoudpour, Matthias Schmidt, Federica Calabrese, Hans Hermann Richnow, Niculina Musat.

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
