## [Decision Letter · Decision Letter 0]

19 Aug 2020

PONE-D-20-22883

High resolution microscopy to evaluate the efficiency of surface sterilization of Zea Mays seeds

PLOS ONE

Dear Dr. Yalda Davoudpour,

Thank you for submitting your manuscript to PLOS ONE. After careful consideration, we feel that it has merit but does not fully meet PLOS ONE’s publication criteria as it currently stands. Therefore, we invite you to submit a revised version of the manuscript that addresses the points raised during the review process.

We look forward to receiving your revised manuscript.

Kind regards,

Yi Cao

Academic Editor

PLOS ONE

Journal Requirements:

Reviewers' comments:

Reviewer's Responses to Questions

**Comments to the Author**

1. Is the manuscript technically sound, and do the data support the conclusions?

Reviewer #1: Yes

Reviewer #2: Partly

2. Has the statistical analysis been performed appropriately and rigorously? 

Reviewer #1: Yes

Reviewer #2: N/A

3. Have the authors made all data underlying the findings in their manuscript fully available?

Reviewer #1: Yes

Reviewer #2: Yes

4. Is the manuscript presented in an intelligible fashion and written in standard English?

Reviewer #1: Yes

Reviewer #2: Yes

5. Review Comments to the Author

Reviewer #1: This report compares a few different sterilization treatment methods of maize seeds. An effective method has been screened out in this report and convincing methodologies to evaluate the effect of sterilization methods on microbial cultivation and germination. Although the number of seeds is small, the results are consistent and promising. Thus, I recommend it to be published with a few minor revisions:

What is the volume of the NaOCl solution during the treatment? 17 uL of Tween 20 added to how much solution?

Explain the controlled medium in Table 1 or the text when mentioning the procedure on pages 6 and 7. The explanation shows up too late.

Fig. 4 labeling should be EtOH-SH-xxx missing SH in two labels.

Reviewer #2: The article, entitled “High resolution microscopy to evaluate the efficiency of surface sterilization of Zea Mays seeds,” by Yalda Davoudpour et al, for publication in POLS ONE, has some interesting results. Their approach, applying optical and scanning electron microscopic imaging as the major tools, has the potential to be developed into routine characterization protocol. There are several flaws and insufficiencies that need to be addressed properly.

Therefore, I recommend publication once those issues (details below) are addressed.

In the methods section, authors mentioned roots were cut to approximately 1 cm for optical microscopy. Also, they show images of grown roots in several images within Figs 2, 3, and 5. It will be helpful to readers if these authors give the diameter of those Petri dishes so one has a sense of actual length of those grown roots.

I am concerned with poor image quality in Figs 2, 3 and 5. It could be the result of a poor production of my copy only. Anyway, even with a cell phone one surely can obtain much higher quality images of this nature. Authors should make sure their original pictures do have adequate contrast to show clearly grown roots.

Fig. 8b needs improvement on how to present it. Its background at lower right quarter is too bright, overwhelming/dominating the lone fluorescent dot in the upper left quarter.

Images in Fig. 10 are very good.

Author’s comments about unable to determine whether microorganisms were on root surface with optical microscopy was a misconception. Optical microscopic imaging is capable do even 3-D sectional imaging, for example with confocal microscopy. Optical lenses form planar images of objects on a plane. At the resolution of their work, there is no doubt that images in Fig. 10 are features on root surface, not internal.

Fig. 11 is actually puzzling.

Those “dots,” deemed detected microorganisms, are not clear et all and are too small. The resolution of HIM should be far better. These authors should show enlarged section of those dots so that their 3-D structure can be elucidated. Images in Fig.11 do not support claims of any detected microorganism. Structures of root hairs are apparent. Red circles are very faint to almost invisible. I only saw a few random and very blurred red dots, no arrows at all.

These authors need to provide details on “cell counting,” both on methods and actual work. By imaging, readers understand how to literally count fluorescent dots in an image. This direct method gives numbers (of cells/microorganisms or else) per unit area. How to convert it to numbers per unit volume is not clear at all. These authors need to have one paragraph on methods. If literally counting “cells” on images, it is more scientific to provide the total number before a conversion to the unit number. In terms of counting, if it is manual, these authors should state so and report on how time-consuming the process actually is.

6. PLOS authors have the option to publish the peer review history of their article (what does this mean?). If published, this will include your full peer review and any attached files.

Reviewer #1: No

Reviewer #2: No

---

## [Author Response · Author response to Decision Letter 0]

23 Sep 2020

Dear Dr. Yi Cao,

We would like to thank you and the reviewers for the valuable comments and suggestions to improve the quality of our manuscript, the time and effort dedicated as well as for the opportunity to submit the revised version. We incorporated the insightful suggestions and comments in the revised version. The changes are marked in the manuscript by yellow highlight. Please find our detailed response to the reviewer’s comments below. One of the critical point in comments of reviewer 2 was concerning the poor image quality/resolution. We have checked all images and find out that during exporting particularly of the multi-panel figures, the resolution of individual images decreased considerably. We have therefore used for the revised version a different software to export final images and keep the initial resolution of the individual images. 

Reviewer’s comments for the manuscript:

Reviewer ≠1:

1. What is the volume of the NaOCl solution during the treatment? 17 uL of Tween 20 added to how much solution?

Author’s response: Thanks for your completely correct point. We added the missing information in the text. Materials and Methods section, Surface sterilization procedures (page 6, line 136).

2. Explain the controlled medium in Table 1 or the text when mentioning the procedure on pages 6 and 7. The explanation shows up too late.

Author’s response: As suggested by the reviewer, we added the explanation about the control medium earlier on page 7, line 153-154. 

3. Fig. 4 labeling should be EtOH-SH-xxx missing SH in two labels.

Author’s response: We added the missing SH to the EtOH-va-st and EtOH-st labels in Fig. 4.

Reviewer ≠2:

1. In the methods section, authors mentioned roots were cut to approximately 1 cm for optical microscopy. Also, they show images of grown roots in several images within Figs 2, 3, and 5. It will be helpful to readers if these authors give the diameter of those Petri dishes so one has a sense of actual length of those grown roots.

Author’s response: We thank the reviewer for the comment. We added the missing scale bars in Fig 2, 3, 5 and the corresponding figures captions (Fig2. page 11, line 239-240) (Fig3. page 12, line 262) (Fig5. page 13, line 295). 

2. I am concerned with poor image quality in Figs 2, 3 and 5. It could be the result of a poor production of my copy only. Anyway, even with a cell phone one surely can obtain much higher quality images of this nature. Authors should make sure their original pictures do have adequate contrast to show clearly grown roots.

Author’s response: We thank the reviewer for observing and pointing out the resolution issues. We have checked and export the multi-panel images using a different software to keep the initial high resolution of individual images.

3. Fig. 8b needs improvement on how to present it. Its background at lower right quarter is too bright, overwhelming/dominating the lone fluorescent dot in the upper left quarter.

Author’s response: We thank to the reviewer for the useful comment. According with this comment we decided to present the fluorescence microscopy images in the revised version of manuscript without any changes in signal to noise ratio (best fit) suggested by the image processing software (ZEN 3.1).

4. Images in Fig. 10 are very good. Author’s comments about unable to determine whether microorganisms were on root surface with optical microscopy was a misconception. Optical microscopic imaging is capable do even 3-D sectional imaging, for example with confocal microscopy. Optical lenses form planar images of objects on a plane. At the resolution of their work, there is no doubt that images in Fig. 10 are features on root surface, not internal.

Author’s response: We thank the reviewer for the valuable explanation. The reviewer comment is completely justified and correct. Therefore, we decided to remove the corresponding statement from the text.

5. Fig. 11 is actually puzzling. Those “dots,” deemed detected microorganisms, are not clear et all and are too small. The resolution of HIM should be far better. These authors should show enlarged section of those dots so that their 3-D structure can be elucidated. Images in Fig.11 do not support claims of any detected microorganism. Structures of root hairs are apparent. Red circles are very faint to almost invisible. I only saw a few random and very blurred red dots, no arrows at all.

Author’s response: We thank to the reviewer for pointing out resolution issues with many of the presented figures. We have checked individually each figure and realized the resolution was decreased during exporting of the multi-panel figures. In the revised version we used another software to export multi-panel images with excellent results for keeping the initial resolution of individual images. In addition, when available we replaced images with close ups for better observation of bacterial cells on the root surface. We made the circles and arrows larger for better recognition and supporting our claim. 

6. These authors need to provide details on “cell counting,” both on methods and actual work. By imaging, readers understand how to literally count fluorescent dots in an image. This direct method gives numbers (of cells/microorganisms or else) per unit area. How to convert it to numbers per unit volume is not clear at all. These authors need to have one paragraph on methods. If literally counting “cells” on images, it is more scientific to provide the total number before a conversion to the unit number. In terms of counting, if it is manual, these authors should state so and report on how time-consuming the process actually is.

Author’s response: We appreciate the reviewer comment. We incorporated the explanation of DAPI cell counting in the material and methods section (page 9, lines 183-194).

---

## [Decision Letter · Decision Letter 1]

30 Oct 2020

High resolution microscopy to evaluate the efficiency of surface sterilization of Zea Mays seeds

PONE-D-20-22883R1

Dear Dr. Davoudpour,

We’re pleased to inform you that your manuscript has been judged scientifically suitable for publication and will be formally accepted for publication once it meets all outstanding technical requirements.

Kind regards,

Yi Cao

Academic Editor

PLOS ONE

Additional Editor Comments (optional):

Reviewers' comments:

Reviewer's Responses to Questions

**Comments to the Author**

1. If the authors have adequately addressed your comments raised in a previous round of review and you feel that this manuscript is now acceptable for publication, you may indicate that here to bypass the “Comments to the Author” section, enter your conflict of interest statement in the “Confidential to Editor” section, and submit your "Accept" recommendation.

Reviewer #1: All comments have been addressed

2. Is the manuscript technically sound, and do the data support the conclusions?

Reviewer #1: Yes

3. Has the statistical analysis been performed appropriately and rigorously? 

Reviewer #1: Yes

4. Have the authors made all data underlying the findings in their manuscript fully available?

Reviewer #1: Yes

5. Is the manuscript presented in an intelligible fashion and written in standard English?

Reviewer #1: Yes

6. Review Comments to the Author

Reviewer #1: (No Response)

7. PLOS authors have the option to publish the peer review history of their article (what does this mean?). If published, this will include your full peer review and any attached files.

Reviewer #1: No

---

## [Editor Report · Acceptance letter]

16 Nov 2020

PONE-D-20-22883R1 

High resolution microscopy to evaluate the efficiency of surface sterilization of Zea Mays seeds 

Dear Dr. Davoudpour:

I'm pleased to inform you that your manuscript has been deemed suitable for publication in PLOS ONE. Congratulations! Your manuscript is now with our production department. 

Kind regards, 

on behalf of

Dr. Yi Cao 

Academic Editor

PLOS ONE